# User Armor: An Extension for AppArmor

Mario Alviano *,†ⓘ and Pierpaolo Sestito †

Department of Mathematics and Computer Science, University of Calabria, 87036 Rende, Italy;
pierpaolo.sestito@outlook.it
* Correspondence: mario.alviano@unical.it
† These authors contributed equally to this work.

**Abstract:** AppArmor is a mandatory access control (MAC) system for Linux based on profiles. It focuses on protecting processes, without differentiating profiles based on the users running the processes themselves. Moreover, it does not implement inheritance mechanisms to simplify the management of profiles and avoid the duplication of rules. This work introduces UserArmor, an extension of AppArmor that overcomes the aforementioned limitations by allowing specific profiles to be associated with users and implementing an inheritance system to reduce complexity, improve reusability, and ensure consistency in security rules. An application to Answer Set Programming is discussed.

**Keywords:** mandatory access control (MAC); profile inheritance; user-based profiles

## 1. Introduction

AppArmor [1] is a security Mandatory Access Control (MAC) framework for Linux [2,3], based on security profiles that apply restriction on the resources available to processes [4,5]. In fact, these profiles define strict policies for accessing files, directories, networks and other critical resources [5,6]. Restrictions are applied at the kernel level, providing a mandatory control independent of application behavior [4]. Even if AppArmor improves multi-user security, it lacks flexibility in managing user-specific rules. Specifically, AppArmor allows profiles to be defined for applications or processes, but it lacks a mechanism to differentiate between different users running the same process. Suck a lack of user-level control makes it complex to ensure security in multi-user scenarios.

**Example 1.** *Let us consider an application that can be executed by two users, namely user1 and user2. The application writes to a file whose name depends on the user running the process. The most restrictive polices that can be implemented in AppArmor for such a scenario must include the following rules:*

```
/var/log/my_confined_app/user1.log rw,
/var/log/my_confined_app/user2.log rw,
```

*It is not possible to further restrict permissions without breaking the application logic, i.e., each user must be able to access its own file via the application. This can lead to the following vulnerabilities: a bug in the application could allow user1 to read or write to the file associated with user2, compromising the confidentiality and integrity of such potentially sensible resources.*

*The proposed extension involves the use of different subprofiles for each user. For the scenario reported above, we would have the following subprofiles:*

```
profile user1 {
    ...
```

```
        /var/log/my_confined_app/user1.log rw,
}
profile user2 {
        ...
        /var/log/my_confined_app/user2.log rw,
}
```

*Therefore, when the application is executed by* `user1`*, the subprofile* `user1` *is enforced, and in this way, access to the* `user2.log` *file is inhibited at kernel level, even in the case of bugs in the application.*

Another limitation of AppArmor is the absence of an inheritance mechanism for profile policies. Even if profiles can be nested, creating a seemingly hierarchical structure, AppArmor subprofiles are designed to define different security rules for subprocesses of an application. The way they are designed, subprofiles do not inherit the rules of the main profile, as it is possible that a subprocess needs a less restrictive security policy to work correctly. Technically, AppArmor implements a notion of abstractions to reuse common rules; such rules are stored in abstraction files and can be reused in other security policies via the `#include` directive. However, abstractions are an all or nothing option: it is not possible to select specific rules within an abstraction, for example, based on the user running the process. Due to the absence of inheritance mechanisms, policy management is cumbersome and prone to errors, especially in complex environments where configurations must be updated frequently or affect multiple users.

**Example 2.** *Consider a Bash Application* `my_confined_app` *that needs the* `cat` *command to read a configuration file and access the network. The application can be run by two users, both sudoers, only one of whom needs administrative capabilities. As in Example* 1*, users must have read and written access to their log files. The use of AppArmor that we propose would therefore need the following file:*

```
#include <tunables/global>
/usr/bin/my_confined_app {
    profile user1 {
        #include <abstractions/base>
        #include <abstractions/bash>
        /usr/bin/my_confined_app r,
        /etc/my_confined_app.conf r,
        /usr/bin/cat ix,
        network inet,
        /var/log/my_confined_app/user1.log rw,
    }

    profile user2 {
        #include <abstractions/base>
        #include <abstractions/bash>
        /usr/bin/my_confined_app r,
        /etc/my_confined_app.conf r,
        /usr/bin/cat ix,
        capability sys_admin,
        network inet,
        /var/log/my_confined_app/user2.log rw,
    }
}
```

*Above, the included abstractions set common permissions for Bash scripts, including access to* `.bash_profile, .bash_rc` *and* `.profile` *files. The file above has several duplicate rules. The*

*extension proposed in this paper provides an inheritance system by which the described scenario can be modeled as follows:*

```
#include <tunables/global>
/usr/bin/my_confined_app {
    #include <abstractions/base>
    #include <abstractions/bash>
    /usr/bin/my_confined_app r,
    /etc/my_confined_app.conf r,
    /usr/bin/cat ix,
    #@selectable{adm}  capability sys_admin,
    #@selectable{net}  network inet,

    profile user1 {
        #@select: adm net
        /var/log/my_confined_app/user1.log rw,
    }

    profile user2 {
        #@select: net
        /var/log/my_confined_app/user2.log rw,
    }
}
```

*In the above policy file, the `#@selectable` and `#@select` directives avoid the duplication of rules. Their usage is detailed in Section 3.*

This paper presents UserArmor, an extension of AppArmor, with the goal of achieving greater granularity in user-level security policy management, and introducing an inheritance system based on tags. The user-level granularity is achieved by structuring AppArmor security policies hierarchically. In the proposed structure, the general application profile serves as the base level, while every user executing the application can be associated with a (nested) subprofile. As shown in Figure 1, in UserArmor, each confined application is associated with a directory containing user subprofiles, each one stored in a separate file to favor the modularity and scalability of the approach. The profiles are then copied into a single file, namely `mappings`, that is included in the security policy via the `#include` directive. In this way, when the security policy of an application `confined_app` is processed, subprofiles are loaded into the kernel and associated with names like `confined_app//user_name`.

UserArmor automates the creation and management of the hierarchical structure of profiles and the activation of the profile associated with the user running an application. Moreover, its inheritance system is designed to eliminate the duplication of rules. The idea is to tag some rules as selectable, and specify in each subprofile which tags to select. Untagged rules, on the other hand, are essential and included in all subprofiles. Figure 2 illustrates the usage of UserArmor for the scenario described in Example 2. UserArmor command-line tools, namely `ua-generate`, `ua-enforce` and `ua-exec`, are described in Section 3.

The remainder of this article is structured as follows. Section 2 introduces the required background on AppArmor, and in particular, the notion of profile and the include directives. Section 3 defines the proposed tag system to reuse AppArmor rules, and the command-line tools introduced by UserArmor are as follows: `ua-generate` to produce skeleton files for user profiles; `ua-enforce` to process user profiles and obtain files understandable by AppArmor; `ua-exec` to execute confined application with the correct profile. Section 4 overviews related works in the literature. Section 5 presents an experiment aimed at measuring the overhead introduced by UserArmor with respect to AppArmor, that is, to exclude denial-of-service

(DoS) vulnerabilities due to excess resource consumption to enforce the user-based policies. The results of our experiment show that the overhead is minimal, and significantly less than the overhead introduced by sandboxing tools such as Bubblewrap. Section 6 reports an application of UserArmor to harden Answer Set Programming solvers [7–11], and in particular, the state-of-the-art solver CLINGO [12].

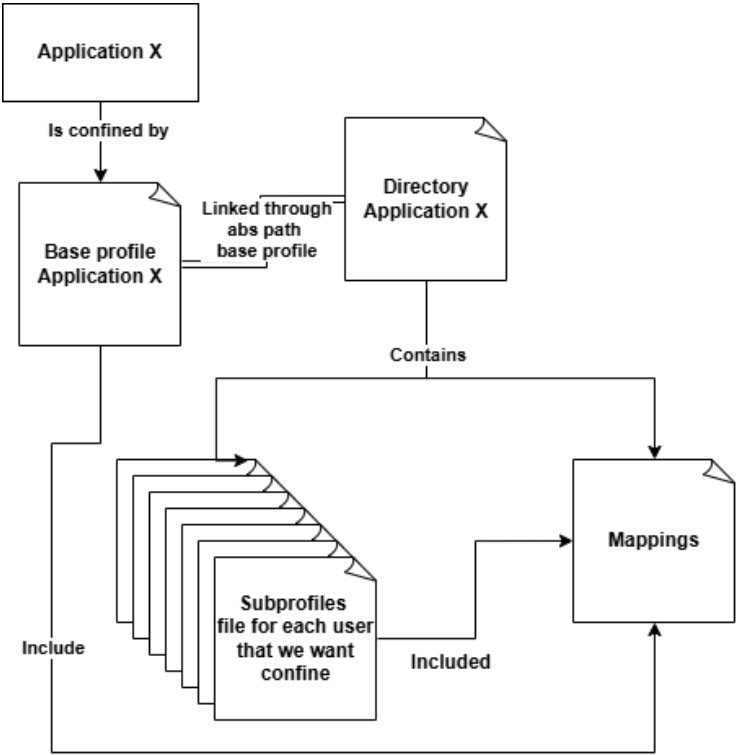

**Figure 1.** Hierarchical structure of UserArmor profiles.

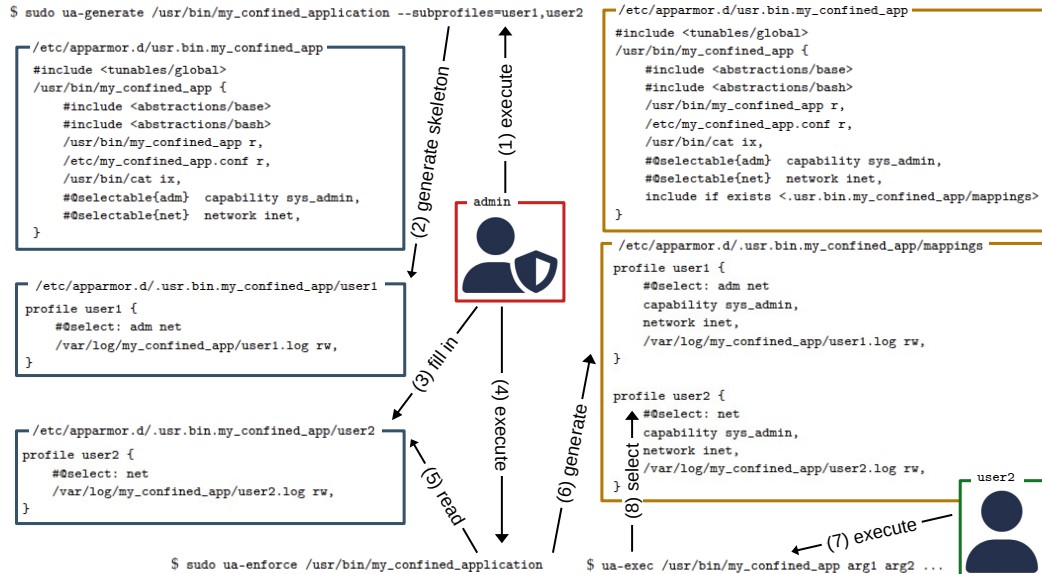

**Figure 2.** UserArmor usage for Example 2. Initially, the administrator executes `ua-generate` to produce skeleton files for user profiles and fills them in with the required permissions (blue boxes). The administrator can take advantage of the UserArmor tag system to reuse common rules. After that, the administrator executes `ua-enforce` to process the written profiles (blue boxes) and generate files that are understandable by AppArmor (orange boxes). Finally, `user2` executes the confined application via `ua-exec`, which selects the profile associated with the user (i.e., `/usr/bin/my_confined_app//user2`).

## 2. Background

In AppArmor, processes can be associated with *profiles* that enables access to system resources. Profile files are conventionally saved in the directory `/etc/apparmor.d/`, with filenames obtained from the absolute path of the executable files (replacing / with .). A profile file has the form

```
profile NAME /ABSOLUTE/PATH {
    RULES
    SUBPROFILES
}
```

where `RULES` is a list of rules (defined next), and `SUBPROFILES` is a possibly empty list of profiles; the keyword `profile` and the `NAME` can be omitted from the main profile. The rules may concern *files*, *capabilities* and *networking*. A rule about files has the form

```
/ABSOLUTE/PATH FLAGS,
```

where `FLAGS` includes one or more flags; flags relevant to this work are `r`, `w` and `x` for reading, writing and executing files; and `ix` for executing files maintaining the current profile. A rule on capabilities has the form

```
capability CAPABILITIES,
```

where `CAPABILITIES` is a capability in the Linux kernel (e.g., `setuid` or `setguid`). A rule on networking has the form

```
network TYPE,
```

where `TYPE` is the type of communication (e.g., `inet` for IPv4 or `inet6` for IPv6). Rules can be preceded by the keyword `deny` to block access (rather than allow access).

AppArmor has several tools for simplifying profile generation and management. The tools most relevant for this work are the following: `aa-genprof` generates a new profile by tracking the execution of an application and asking the user to confirm or deny the accesses requested by the application; `aa-logprof` analyzes the audit log files and helps improve an existing profile by suggesting new rules based on recorded events; `apparmor_parser` loads and reloads existing profiles into the kernel, verifies and compiles profiles transforming them into a kernel format; `aa-complain` activates the logging of a profile, without enforcing its restrictions; `aa-enforce` applies restrictions (and logging) of a profile; `aa-disable` disallows restrictions and logging of a profile.

Finally, AppArmor allows the reuse of common rules through abstraction files, which can be included with the directive

```
#include <file>
```

where `file` is the path of the abstraction file relative to the directory `/etc/apparmor.d/`. If the file does not exist, AppArmor raises an error. Alternatively, it is possible to use

```
include if exists <file>
```

to include the file under the condition of its existence.

## 3. User Armor

The tag system uses a comment-based syntax, described below. Rules (and blocks) are associated with aliases (i.e., identifying strings).

- Base rules. Untagged rules are considered essential and are automatically inherited by all subprofiles.
- Selectable rules. Rules starting with a tag

  ```
  #@selectable{ALIAS}
  ```

are not inherited automatically (and are not part of the main profile), but they can be included in the subprofiles via their `ALIAS`.

- Selectable blocks. A syntax similar to the previous one can be applied to rule blocks as follows:

  ```
  #@selectable{ALIAS}
  #    RULES
  #@end
  ```

- Removable rules. Rules ending with a tag

  ```
  #@removable{ALIAS}
  ```

  are inherited in subprofiles unless explicitly removed by their `ALIAS`. Removable rules enable the possibility to have a more relaxed base policy (which can ease the gradual integration of UserArmor, but this is discouraged on a stable environment). Note that there is no notion of a removable block.

- Subprofile inheritance. Subprofiles can select rules and blocks using the following tag:

  ```
  #@select: ALIASES
  ```

  where `ALIASES` is a space-separated list of aliases. Selected rules and blocks are added to the basic rules. Untagged rules are added at the beginning of each subprofile, unless their alias is listed in the following tag:

  ```
  #@remove: ALIASES
  ```

The tag system is applied to profile files within a directory associated with the confined application. The directory contains a profile file for each involved user and a `mappings` file that includes all the other files in the directory. The `mappings` file can then be included in the profile file in `/etc/apparmor.d`. In this way, AppArmor loads subprofiles into the kernel with name `confined_app//user_name`, and UserArmor can easily apply the correct profile by identifying the user who is running the application.

**Example 3** (Continuing Example 2). *The scenario depicted in the introduction can be modeled by the following profile file `/etc/apparmor.d/usr.bin.my_confined_app`:*

```
#include <tunables/global>
/usr/bin/my_confined_app {
    #include <abstractions/base>
    #include <abstractions/bash>
    /usr/bin/my_confined_app r,
    /etc/my_confined_app.conf r,
    /usr/bin/cat ix,
    #@selectable{adm}  capability sys_admin,
    #@selectable{net}  network inet,
    include if exists <.usr.bin.my_confined_app/mappings>
}
```

*Additional files are stored in the directory `/etc/apparmor.d/.usr.bin.my_confined_app`, as follows:*

- *`user1`, associated with the first user, with the following content:*

  ```
  profile user1 {
      #@select: adm net
      /var/log/my_confined_app/user1.log rw,
  }
  ```

- *`user2`, associated with the second user, with the following content:*

  ```
  profile user2 {
  ```

```
            #@select: net
            /var/log/my_confined_app/user2.log rw,
       }
```

- *mappings, including the above files with the expansion of the selected permissions, as follows:*

```
       profile user1 {
           #@select: adm net
           capability sys_admin,
           network inet,
           /var/log/my_confined_app/user1.log rw,
       }

       profile user2 {
           #@select: adm
           network inet,
           /var/log/my_confined_app/user2.log rw,
       }
```

*UserArmor simplifies the creation of the above files by automatically adding the* `include if exists <.usr.bin.my_confined_app/mappings>` *directive in the profile of the confined application and generating the content of the file* `/etc/apparmor.d/.usr.bin.my_confined_app/mappings`.

UserArmor comprises the following three command-line utilities: `ua-generate`, `ua-enforce`, and `ua-exec`. These tools simplify the process of defining and enforcing user-specific security policies, ensuring that applications run with the appropriate restrictions based on the user executing them. The `ua-generate` command is the starting point for setting up UserArmor. Given the absolute path of an executable and a comma-separated list of users, it automatically creates the necessary directory structure and generates a subprofile file for each specified user. If a subprofile already exists, it remains unchanged, preventing unintended overwrites. Since this command modifies AppArmor profiles, it must be executed with superuser privileges. Once the user-specific profiles are in place, `ua-enforce` takes over to integrate them into the main AppArmor profile of the executable. Given the absolute path of the executable, it ensures that the AppArmor profile contains the necessary `include if exists` directive, allowing the system to conditionally load the user-specific subprofiles. Such subprofiles are collected in the `mappings` file, which is also generated by `ua-enforce` by using the tag system defined in Section 3. Like `ua-generate`, this command requires root privileges, as it modifies the security policies of AppArmor. The final piece of the puzzle is `ua-exec`, which ensures that when an executable is run, it is confined under the correct UserArmor subprofile. Unlike the previous two utilities, `0|ua-exec|` can be run by any user. It works by identifying the current user, selecting the corresponding subprofile, and executing the application via `aa-exec` to enforce the correct restrictions. For security reasons, `aa-exec` is expected to be executable only by the root and members of the `userarmor` group, preventing unauthorized users from bypassing confinement. This way, users in the `userarmor` group are correctly recognized and associated with their subprofile, while users that are not members of the `userarmor` group will be forced to use the base profile (i.e., the standard model adopted by AppArmor).

**Example 4** (Continuing Example 3)**.** *The starting point is the file defining the base profile. Given the file* `/etc/apparmor.d/usr.bin.my_confined_app` *from Example 3, with or without the* `include if exists`*, UserArmor can generate the directory structure with the following command:*

```
$ sudo ua-generate /usr/bin/my_confined_application --subprofiles=user1,user2
```

*The user profiles are initially empty (unless the files already exist), and the administrator can populate them with the content reported in Example 3. After that, the administrator can issue the `ua-enforce` command to collect the subprofiles in the `mappings` file and to add the `include if exists` directive to the `/etc/apparmor.d/usr.bin.my_confined_app` file, if not already present, as follows:*

```
$ sudo ua-enforce /usr/bin/my_confined_application
```

*With the above command, UserArmor also takes care of interacting with AppArmor in order to enforce the provided security policy. In order to execute `my_confined_app`, `user1` issues the following command:*

```
$ ua-exec /usr/bin/my_confined_app arg1 arg2 ...
```

*Note that bypassing `ua-exec` with the command*

```
$ my_confined_app arg1 arg2 ...
```

*would run the application with the base profile, hence, with no access to the `/var/log/my_confined_app/user1.log` file.*

## 4. Literature Review

The closest work to our proposal is Paranoid Penguin [5], which introduces an extension of AppArmor primarily aimed at improving usability. The main contribution of Paranoid Penguin is the integration of a graphical user interface (GUI) to facilitate the creation, modification and management of AppArmor profiles. This approach makes AppArmor more accessible to users who may not be familiar with its command-line tools, lowering the barrier to entry for system administrators and security practitioners. The GUI provides an intuitive way to define and enforce security policies without requiring deep knowledge of the syntax used by AppArmor, reducing the risk of misconfigurations that could compromise security. While Paranoid Penguin enhances the usability of AppArmor, it does not extend or modify the underlying security enforcement model. That is, the access control mechanism itself remains unchanged, and security policies are still applied in the same way as in standard AppArmor. Stated differently, the framework does not introduce new functionalities for improving security beyond profile simplification.

In contrast, UserArmor goes beyond usability improvements by introducing two key enhancements to the core security model of AppArmor. First, UserArmor allows binding security profiles to specific users, ensuring fine-grained access control in multi-user environments. Traditional AppArmor applies profiles at the application level, meaning that all users executing the same process are subject to the same restrictions. This limitation can lead to security gaps, as different users may require different levels of access to resources. UserArmor addresses this by associating distinct subprofiles with individual users, ensuring that permissions are customized per user rather than being uniformly applied to all instances of an application. Second, UserArmor introduces an inheritance system for policy rules, which improves policy management and scalability. In standard AppArmor, profiles must explicitly define all required permissions, leading to redundancy when multiple profiles share common rules. While AppArmor provides a mechanism called abstractions to group reusable rules, these operate by an all-or-nothing inclusion, meaning administrators cannot selectively inherit only specific rules from an abstraction. UserArmor solves this problem by introducing a tag-based inheritance mechanism, where profiles can inherit only the necessary rules, reducing duplication and simplifying policy maintenance. By addressing these limitations, UserArmor extends the capabilities of AppArmor beyond usability enhancements and introduces structural improvements that make it more suitable for multi-user environments and dynamic security policies. Stated

differently, while Paranoid Penguin provides a more user-friendly interface for AppArmor, UserArmor fundamentally improves its security model by enabling user-specific policy enforcement and hierarchical policy inheritance, making it a more flexible and scalable solution for access control in modern Linux systems.

A broader area of research focuses on dynamic profiling to enhance security at runtime. These approaches monitor application behavior and adaptively adjust permissions, allowing for more flexible and responsive access control. Unlike traditional security policies, which are static and predefined, dynamic profiling mechanisms observe how applications interact with system resources and adjust security policies accordingly to mitigate risks. One branch of research explores cloud-based profile generation, as seen in [13,14], where profile management is offloaded to Cloud Services. This approach offers greater flexibility and scalability, as security policies can be updated dynamically based on centralized monitoring and threat detection. On the other hand, some solutions are designed to run directly on the operating system, ensuring real-time policy enforcement at the local level [15]. These techniques provide a more fine-grained and immediate reaction to potential security threats, without requiring cloud infrastructure.

Among the various dynamic profiling approaches, containerized environments, particularly Docker-based infrastructures, have received significant attention due to their unique security challenges. Since containers package applications with their dependencies, they are vulnerable to container breakout attacks and misconfigurations that may lead to privilege escalation. Research works such as [16,17] propose automated profile generation techniques for containers, leveraging Auditd and SystemTap to analyze the required permissions dynamically. These solutions have been further integrated into a unified framework [13], which combines the strengths of both methods, ensuring a more robust security model. Another significant advancement is given by [18], which introduces dynamic runtime updates for container security policies. This mechanism allows security rules to be adjusted on the fly as the application state evolves, reducing the need for manual intervention. With the widespread adoption of container orchestrators like Kubernetes, Ref. [19] extends these profiling techniques to automatically generate AppArmor profiles for distributed containers in Kubernetes clusters. This automation streamlines security management and reduces misconfiguration risks in large-scale cloud environments. These studies share the overarching goal of optimizing security enforcement—either by introducing new functionalities or enhancing existing mechanisms. UserArmor aligns with these objectives, but instead of focusing on runtime profiling, it introduces structural improvements to AppArmor by enabling user-specific profile enforcement and hierarchical policy inheritance. While dynamic profiling aims to enhance security by adapting to application behavior, UserArmor provides a systematic and scalable approach to user-aware security policies, ensuring multi-user protection and reducing policy redundancy. Furthermore, the ability of UserArmor to limit process resource usage (e.g., CPU, memory, open files) makes it a complementary solution to dynamic profiling methods, contributing to a more robust and flexible security ecosystem.

Finally, we acknowledge that Linux already provides Role-Based Access Control (RBAC) via SELinux. However, SELinux is significantly more complex than AppArmor and is typically used with predefined policies rather than custom configurations per user [6,20,21]. Organizations that deploy SELinux often rely on standard policies without distinguishing between different users running the same application, because customizing SELinux requires deep knowledge of the security contexts, domains, and type enforcement. UserArmor, on the other hand, complements the simpler AppArmor security mechanisms by introducing a finer-grained, user-specific confinement model, ensuring that each user receives a tailored security profile without modifying global policies.

## 5. Experiment

In order to assess UserArmor empirically, we designed a benchmark comprising the security policies in the examples given in the Introduction. Our benchmark comprises a very simple Bash script, as follows:

```
cat /etc/my_confined_app.conf  >> /var/log/my_confined_app/$USER.log
```

The above script needs to read access to the file `/etc/my_confined_app.conf`, and write access to the file `/var/log/my_confined_app/$USER.log`. Note that the name of the log file is determined using the environment variable `USER`, the value of which is usually the username of the logged-in user. Hence, in the expected usage, if `user1` executes the above Bash script, the log file is `/var/log/my_confined_app/user1.log`. On the other hand, this is a weak assumption, and `user1` can easily reassign the environment variable to corrupt the file associated with `user2`, i.e., `/var/log/my_confined_app/user2.log`. We measure the execution time over repeated executions of the script. The experiment is run on a 13th Gen Intel(R) Core(TM) i7-1360P @ 2.2 GHz CPU with 32 GB RAM, and its aim is to determine the overhead introduced by UserArmor.

The overall results are shown in Figure 3. We observe that AppArmor introduces no overhead, while Bubblewrap is more expensive. The performance of UserArmor is between the following two alternatives, as expected: it works on top of AppArmor, so it cannot be faster than AppArmor, but anyhow less expensive than Bubblewrap. We observe that the time measured for UserArmor are closer to those obtained by AppArmor than to the time needed by Bubblewrap, which is a positive result.

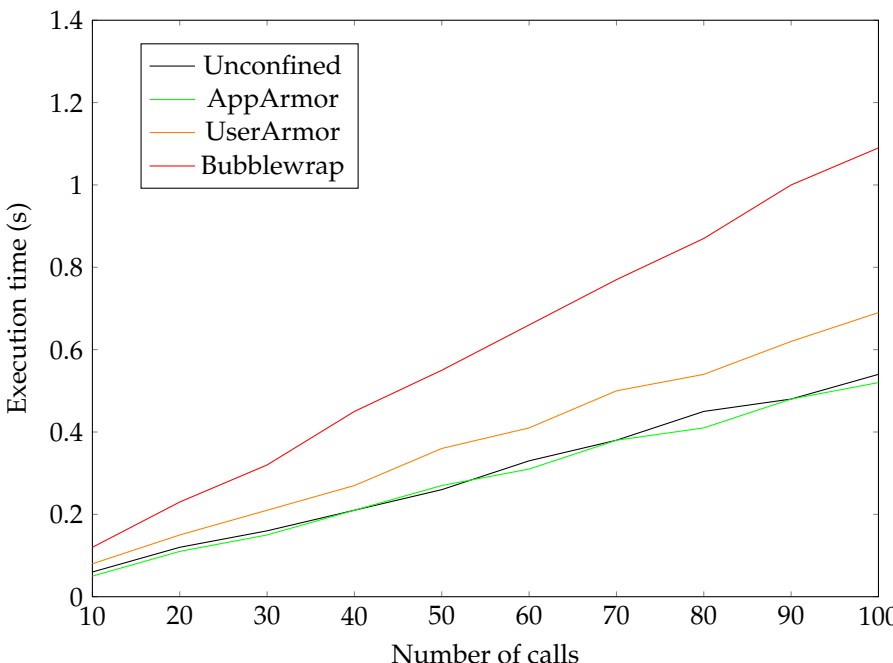

**Figure 3.** Execution time for running the Bash script in our benchmark under different security policies.

## 6. Application: Hardening Answer Set Programming Solvers

Answer Set Programming (ASP) is a declarative programming paradigm designed for solving complex combinatorial problems [22–24]. It is widely used in artificial intelligence, knowledge representation and automated reasoning [25–28]. Without going into much details, ASP programs comprise logic rules and are associated with zero or more stable models, that is, first-order models satisfying an additional stability condition designed to interpret default negation.

**Example 5.** *Let us consider the Hamiltonian Path problem:*

> *Given a directed graph G, a starting node s and a target node t, we need to find a path from s to t that visits all nodes of G exactly once.*

*The graph shown in Figure 4 can be represented by the following ASP:*

```
start("A").  target("G").
node("A").  link("A","B").  link("A","C").
node("B").  link("B","A").  link("B","C").  link("B","D").
node("C").  link("C","A").  link("C","D").
node("D").  link("D","B").  link("D","C").  link("D","E").
node("E").  link("E","D").  link("E","F").  link("E","H").
node("F").  link("F","E").  link("F","G").
node("H").  link("H","F").
node("G").  link("G","F").  link("G","H").
```

*We couple the above facts with rules addressing the Hamiltonian Path problem as follows:*

```
reach(X) :- start(X).
reach(Y) :- next(X,Y).

{next(X,Y) : link(X,Y)} = 1 :- reach(X), not target(X).
:- next(X,Y), next(Z,Y), X < Z.
:- node(X), not reach(X).
```

*Notably, the first two rules above define the reached nodes (`reach/1`) from the starting node (`start/1`), following the selected links (`next/2`). The links are selected by the third rule, which is also called the choice rule; for every reached node not being the target node, exactly one of its outgoing link is selected. The last two rules above, which are also called constraints, ensure that no node has two selected incoming links and that every node is reached. Stable models of the above program represent the Hamiltonian Paths of the given graph; in this case, the unique stable model extends the input facts with the following facts:*

```
reach("A").  reach("B").  reach("C").  reach("D").
reach("E").  reach("F").  reach("H").  reach("G").
next("A","B").  next("B","C").  next("C","D").  next("D","E").
next("E","H").  next("F","G").  next("H","F").
```

*The example can be run in the browser using the following ASP Chef [29,30] recipe: https://asp-chef.alviano.net/s/hamiltonian-path@algorithm2025 (accessed on 26 February 2025). A graphical representation of the computed Hamiltonian Path is given in Figure 4.*

The state-of-the-art solver CLINGO [12] is one of the most popular ASP solvers, offering a powerful inference engine that can be extended with Python v3.12 and Lua v5.4 scripts to enhance reasoning capabilities. Other frequently used ASP solvers are DLV [31], SMODELS [32], CMODELS [33], IDP [34] and WASP [35]. ASP can be used to develop web-based AI applications, and this is a common scenario for researchers aiming at showcasing their AI-powered tools that solve hard combinatorial tasks using ASP [36–38]. They provide a web app with a user-friendly interface (e.g., a textarea for input), where an ASP solver is executed in the backend server to process queries and return computed results.

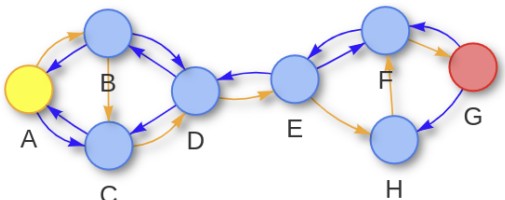

**Figure 4.** Graph used in Example 5. A Hamiltonian Path from A to G is shown using yellow arrows. Links not being part of the computed Hamiltonian Path are shown in blue.

While this approach is functional, it introduces serious security vulnerabilities. CLINGO supports @-terms, which allow interpreted functions to be evaluated dynamically. These functions can be defined in either Lua or Python, meaning that CLINGO can execute arbitrary Python or Lua code. This feature is useful for extending the solver, but it also exposes the system to remote code execution (RCE) vulnerabilities. Malicious users could input a crafted ASP program containing an @-term that executes system commands, leading to arbitrary code execution on the backend server. For example, a user could submit an ASP program like the following:

```python
#script(python)
import~subprocess

def rce(cmd):
    return subprocess.check_output(["sh", "-c", cmd.string], text=True)

#end.

out(@rce("whoami")).
```

Or, in Lua, as follows:

```lua
#script(lua)

function rce(cmd)
    local f = assert(io.popen(cmd.string, 'r'))
    local output = f:read('*a')
    f:close()
    return output
end

#end.

out(@rce("whoami")).
```

The above programs can be easily adapted to delete critical files, exfiltrate data, or install malware on the backend server, completely compromising its security.

A robust solution is to contain the execution of CLINGO by restricting the permissions of the process running the web app. This is where UserArmor comes into play. By defining a UserArmor security profile for `www-data` (or the user running the web service), we can limit CLINGO permissions to the absolute minimum required. More specifically, we can enforce the following restrictions:

- No filesystem access: Prevents CLINGO from reading or modifying system files.
- No network access: Blocks CLINGO from making outbound requests, preventing data exfiltration or malware downloads.
- Minimal execution environment: Restricts process execution, preventing the use of external system tools.

A UserArmor profile for `www-data` could take the following form:

```
profile www-data {
    # Deny access to the entire filesystem except for necessary paths
    deny / rwx,
    /usr/lib/** rm,

    # Allow read access only to the directory where ASP encodings are stored
    /var/www/clingo_input/** r,

    # No network access
    deny network inet,
    deny network inet6,

    # Deny execution of system commands
    deny capability sys_admin,
    deny capability setuid,
    deny capability setgid,
}
```

With this policy, even if an attacker injects Python or Lua code, CLINGO cannot access critical files, execute system commands, or connect to the internet, effectively neutralizing the RCE attack.

Without UserArmor, the only alternative is to apply AppArmor restrictions to CLINGO globally. This means that every instance of CLINGO on the backend server would be restricted, including those run by legitimate users (e.g., via SSH). This is problematic because some users may need unrestricted access to CLINGO. For example, researchers or developers using CLINGO from the terminal might rely on @-terms for legitimate purposes, such as reading external files and making network requests to fetch data. By applying a global AppArmor profile to CLINGO, all users on the system would be restricted in the same way, even those who should have full access.

A possible alternative to UserArmor is to execute CLINGO inside a sandbox, such as Bubblewrap (`bwrap`). Bubblewrap is a lightweight user-space sandboxing tool that can create isolated environments for applications. For example, CLINGO could be executed inside a Bubblewrap sandbox as follows:

```
$ bwrap --unshare-net --ro-bind /usr/bin/clingo /usr/bin/clingo \
    --ro-bind /usr/lib /usr/lib --ro-bind /lib /lib --ro-bind /lib64 /lib64 \
    --ro-bind /var/www/clingo_input /input  /usr/bin/clingo /input/encoding.lp
```

This setup prevents network access (`--unshare-net`) and restricts filesystem access (`--ro-bind` for read-only mounts). However, sandboxing introduces more overhead than AppArmor/UserArmor (every CLINGO execution requires setting up a new isolated environment) and is more difficult to manage (running a sandbox requires an invasive change in how CLINGO is invoked, which may not be practical in existing systems).

We provide scripts showcasing the above scenarios as follows:

- Unconfined execution of CLINGO: This script configures a web app vulnerable to RCE and subject to system takeover.
- Confined execution of CLINGO via AppArmor: This script configures a web app that is not vulnerable to RCE, but SSH users are subject to the same restriction of the web app.
- Confined execution of CLINGO via UserArmor: This script configures a web app that is not vulnerable to RCE, and SSH users are free to use CLINGO as they would normally do.

- Sand-boxed execution of CLINGO via Bubblewrap: This script configure a web app that is not vulnerable to RCE but requires a new isolated for each execution of CLINGO.

The script files are available online (https://github.com/pierpaolosestito-dev/ASPArmor; accessed on 13 February 2025). We ran a second experiment using the aforementioned scripts on a 13th Gen Intel(R) Core(TM) i7-1360P @ 2.2 GHz CPU with 32 GB RAM. In this experiment, we repeatedly call CLINGO to produce 10 thousand integers with the program

`number(1..10,000)`.

We opted for such a simple, deterministic program to obtain a stable performance of CLINGO. As shown in Figure 5, the results confirm the observations from the previous section as follows: UserArmor introduces a negligible overhead, and its performance is superior to Bubblewrap.

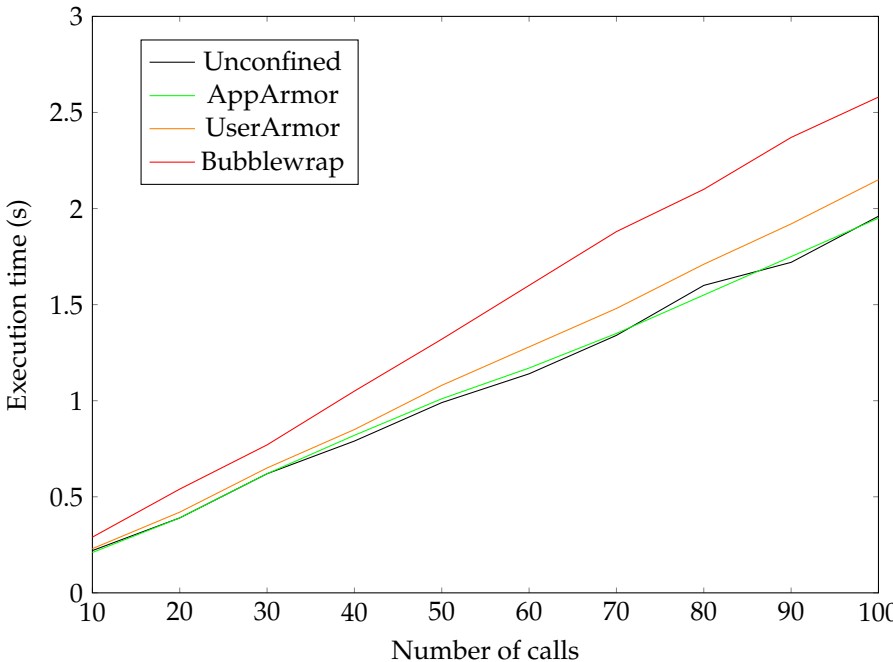

**Figure 5.** Execution time of repeated calls to CLINGO under different security policy mechanisms.

To sum up, hardening CLINGO in a web-based environment is crucial due to its support for @-terms, which can lead to RCE vulnerabilities. While sandboxing tools like Bubblewrap can provide isolation, they introduce overhead and require modifications to the execution workflow. UserArmor, on the other hand, offers a more lightweight and kernel-enforced approach, allowing administrators to restrict CLINGO permissions only for the web service user (`www-data`), without affecting other users who might need full access. For completeness, we also mention the possibility to run CLINGO (and similar engines like MiniZinc [39] and Nemo [40,41]) directly in the browser, by relying on their WebAssembly versions; this is the case, for example, of ASP Chef [29,30], which provides a low-code programming environment powered by ASP and integrating several languages and frameworks [42,43].

## 7. Conclusions

In this paper, we introduced UserArmor, an extension of AppArmor that enhances security policy management by incorporating user-level granularity and inheritance mechanisms. Our approach addresses the following two major limitations of AppArmor: the inability to differentiate permissions for different users running the same process and the lack of an efficient inheritance system for security policies. By structuring security profiles hierarchically and introducing selectable rules via tagging, UserArmor allows administra-

tors to enforce user-specific security constraints while minimizing policy duplication. This improves security, maintainability, and scalability, especially in multi-user environments where different users require distinct levels of access. Our implementation includes a set of CLI tools to facilitate profile management, ensuring that UserArmor is practical and easy to use. Performance evaluation shows that the additional overhead introduced is negligible. Future work on UserArmor will consider the definition of additional tools to support common operations with user accounts, like user renaming (profile files must be renamed accordingly) and deletion (the associated profile files can be deleted as well).

**Author Contributions:** Conceptualization, P.S.; methodology, M.A.; software, P.S.; validation, P.S.; formal analysis, M.A.; investigation, P.S.; resources, P.S.; data curation, P.S.; writing—original draft preparation, P.S.; writing—review and editing, M.A.; visualization, P.S.; supervision, M.A.; project administration, M.A.; funding acquisition, M.A. All authors have read and agreed to the published version of the manuscript.

**Funding:** This work was supported by the Italian Ministry of University and Research (MUR) under PRIN project PRODE "Probabilistic declarative process mining", CUP H53D23003420006, under PNRR project FAIR "Future AI Research", CUP H23C22000860006, under PNRR project Tech4You "Technologies for climate change adaptation and quality of life improvement", CUP H23C22000370006, and under PNRR project SERICS "SEcurity and RIghts in the CyberSpace", CUP H73C22000880001; by the Italian Ministry of Health (MSAL) under POS projects CAL.HUB.RIA (CUP H53C22000800006) and RADIOAMICA (CUP H53C22000650006); by the Italian Ministry of Enterprises and Made in Italy under project STROKE 5.0 (CUP B29J23000430005); under PN RIC project ASVIN "Assistente Virtuale Intelligente di Negozio" (CUP B29J24000200005); and by the LAIA lab (part of the SILA labs). Mario Alviano is member of Gruppo Nazionale Calcolo Scientifico-Istituto Nazionale di Alta Matematica (GNCS-INdAM).

**Data Availability Statement:** Source code and data used in this article are available online at https://github.com/pierpaolosestito-dev/ASPArmor.

**Conflicts of Interest:** The author declare no conflict of interest. The funders had no role in the design of the study; in the collection, analyses, or interpretation of data; in the writing of the manuscript; or in the decision to publish the results.

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
