# Peer review of "User Armor: An Extension for AppArmor"

_algorithms, doi:10.3390/a18040185_

Round 1

Reviewer 1 Report

Comments and Suggestions for Authors

The paper presents an extension of Unix's mandatory access control system, that is claimed to allow more precise and granular management of access rights while simplifying it through profile inheritance. There are the following major drawbacks.

1. The paper does not present a research. It presents a solution to some problem. One aspect of the solution (execution time) is validated through experiment and the proposed method seems better in this aspect. However, other aspects are not studied. This includes but is not limited to: (1) convenience to the user (i.e. convenience to system administrator, who will user UserArmor instead of AppArmor), (2) compatibility with other administrator SW, that is likely to employ AppArmor, but is not aware of UserArmor, (3) migration, or deletion, or other operations with accounts, that can produce orphan files or unresolved links due to inheritance. Specifically for item (1): currently Unix administrators should keep in mind many concepts and SW rules for doing their job successfully. Would they be happy with introducing a new complex concept of UserArmor instead of relatively simple AppArmor? These questions should be researched and discussed.

2. The structure of the paper is strange. 'Related work' section is normally placed closer to beginning, rather than in the end. Placing lengthy examples and image in the introduction is also not good. It is better to have these things in 'background' or 'related work' sections.

3. The paper is submitted to 'Algorithms' journal. But in fact no algorithm or method is presented. Some new rules and SW utilities are shown. No algorithm in formal of "step 1, step 2...", no block scheme. I think the paper is not suitable for 'Algorithms' and should be readdressed to some journal on computer security.

Minor issues.

1. Line 21: 'police'->'policy"

2. Figure 1 caption should be in English.

Author Response

| The paper presents an extension of Unix's mandatory access control system, that is claimed to allow more precise and granular management of access rights while simplifying it through profile inheritance. There are the following major drawbacks.

| 1. The paper does not present a research. It presents a solution to some problem. One aspect of the solution (execution time) is validated through experiment and the proposed method seems better in this aspect. However, other aspects are not studied. This includes but is not limited to: (1) convenience to the user (i.e. convenience to system administrator, who will user UserArmor instead of AppArmor), (2) compatibility with other administrator SW, that is likely to employ AppArmor, but is not aware of UserArmor, (3) migration, or deletion, or other operations with accounts, that can produce orphan files or unresolved links due to inheritance. Specifically for item (1): currently Unix administrators should keep in mind many concepts and SW rules for doing their job successfully. Would they be happy with introducing a new complex concept of UserArmor instead of relatively simple AppArmor? These questions should be researched and discussed.

We commented on these aspects at the end of Literature Review:

Finally, we acknowledge that Linux already provides Role-Based Access Control (RBAC) via SELinux.
However, SELinux is significantly more complex than AppArmor and is typically used with predefined policies rather than custom configurations per user.
Organizations that deploy SELinux often rely on standard policies without distinguishing between different users running the same application because customizing SELinux requires deep knowledge of security contexts, domains, and type enforcement.
UserArmor, on the other hand, complements the simpler AppArmor security mechanisms by introducing a finer-grained, user-specific confinement model, ensuring that each user receives a tailored security profile without modifying global policies.

Specifically for (3) we added a future work in the conclusion:

Future work on UserArmor will consider the definition of additional tools to support common operations with user accounts, like user renaming (profile files must be renamed accordingly) and deletion (the associated profile files can be deleted as well).

| 2. The structure of the paper is strange. 'Related work' section is normally placed closer to beginning, rather than in the end. Placing lengthy examples and image in the introduction is also not good. It is better to have these things in 'background' or 'related work' sections.

We restructured the paper as suggested, and moved section 6 (now Literature Review) before the empirical work.

| 3. The paper is submitted to 'Algorithms' journal. But in fact no algorithm or method is presented. Some new rules and SW utilities are shown. No algorithm in formal of "step 1, step 2...", no block scheme. I think the paper is not suitable for 'Algorithms' and should be readdressed to some journal on computer security.

We added an infographic with the main steps to use UserArmor (Figure 2, in the introduction as suggested by Rev#2).

| Minor issues.

| 1. Line 21: 'police'->'policy"

Fixed.

| 2. Figure 1 caption should be in English.

Fixed.

Reviewer 2 Report

Comments and Suggestions for Authors

Dear authors,

The study needs major changes as displayed bellow:

  • Caption figure 1 is in Italian. Please, use plain English
  • Introduction section needs to introduce the design of the study, either graphically, or textually.
  • No need of repeating the sentence ‘AppArmor (Application Armor) is a Mandatory Access Control (MAC) framework for Linux’.
  • Lack of rationale linkage between sections, resulting in missing vein along the text overall
  • This intends to be a scientific article and thus it is mandatory to include more literature review and ensuing references.
  • Very long paragraphs that make the text confuse. Needs rewritten.
  • The section 6 should come up earlier, along with section 2 ‘Background’ and before the empirical work.

Finally, there are some challenges that come up unnoticed and should be referenced in the main text, in particular:

  • Redundancy with other security processes, as Linux already presents the Role-Based Access. Firms that are already adopting SELinux and AppArmor may see redundant to use the UserArmor.
  • The likely increased overhead and complexity, leading to possible mistakes in different environs

Best regards

Comments on the Quality of English Language

The english needs minor proofreading

Author Response

| Dear authors,

| The study needs major changes as displayed bellow:

Thank you for your comments. Below is a pointwise response.

| Caption figure 1 is in Italian. Please, use plain English

Oops! We deserve Espresso Macchiato.

Changed to "Figure 1. Hierarchical structure of UserArmor profiles".

| Introduction section needs to introduce the design of the study, either graphically, or textually.

We added Figure 2 and its description.

| No need of repeating the sentence ‘AppArmor (Application Armor) is a Mandatory Access Control (MAC) framework for Linux’.

Removed.

| Lack of rationale linkage between sections, resulting in missing vein along the text overall
| This intends to be a scientific article and thus it is mandatory to include more literature review and ensuing references.
| Very long paragraphs that make the text confuse. Needs rewritten.
| The section 6 should come up earlier, along with section 2 ‘Background’ and before the empirical work.

We restructured the paper as suggested, and moved section 6 (now Literature Review) before the empirical work.

We added more references. We hope the submission system is including the references (in the past we add problems like this). Here is a PDF of our submission:

https://www.dropbox.com/scl/fi/3x1ebwatw6ux51b2drnvo/User_Armor___Algorithms-1.pdf?rlkey=rg0jn9vxnj4lq6tqjvd9mkt9d&dl=1

| Finally, there are some challenges that come up unnoticed and should be referenced in the main text, in particular:

| Redundancy with other security processes, as Linux already presents the Role-Based Access. Firms that are already adopting SELinux and AppArmor may see redundant to use the UserArmor.
| The likely increased overhead and complexity, leading to possible mistakes in different environs

We commented on this aspect at the end of Literature Review:

Finally, we acknowledge that Linux already provides Role-Based Access Control (RBAC) via SELinux.
However, SELinux is significantly more complex than AppArmor and is typically used with predefined policies rather than custom configurations per user.
Organizations that deploy SELinux often rely on standard policies without distinguishing between different users running the same application because customizing SELinux requires deep knowledge of security contexts, domains, and type enforcement.
UserArmor, on the other hand, complements the simpler AppArmor security mechanisms by introducing a finer-grained, user-specific confinement model, ensuring that each user receives a tailored security profile without modifying global policies.

Reviewer 3 Report

Comments and Suggestions for Authors

This manuscript is important because it brings additional security policies in configuring user profiles. To make this manuscript as good as possible, you should keep the following remarks in mind.

On line 21, please reformulate "The most restrictive police that can be implemented", it is a confusing formulation (police? I think is about policies....).
It is good to have a test for the existence of the .bash_profile and .bash_login files, which are read before the .profile file, in order not to allow other vulnerabilities.
The general rule of writing being in English, please correct "Figure 1. Struttura gerarchica dei profili" to "Figure 1. Hierarchical structure of profiles" (or similar).
On line 289 it appears written "32 GiB RAM". Most likely it is "32 GB RAM". On line 404 the same.
About the "Bubblewrap" procedure, no details are provided at the first appearances (the first reference is to Figure 2 and line 292), but only at line 380. It is good that any name (consecrated or not) be presented at the first invocation.
For CLINGO (some details about it), other possibilities can be mentioned and, why not, it would have been good to compare them with other ASP solutions.

Figure 2 and Figure 3 are fine, but it's better to do more testing. For example, can UserArmor (since it derives from AppArmor which is implemented at the kernel level) be exploited by DoS? Vulnerabilities, depending on the kernel version, have allowed attackers to bypass AppArmor's protections.

Author Response

| This manuscript is important because it brings additional security policies in configuring user profiles. To make this manuscript as good as possible, you should keep the following remarks in mind.

Thank you for your comments. Below is a pointwise response.

| On line 21, please reformulate "The most restrictive police that can be implemented", it is a confusing formulation (police? I think is about policies....).

Done.

| It is good to have a test for the existence of the .bash_profile and .bash_login files, which are read before the .profile file, in order not to allow other vulnerabilities.

We added a note on Example 2:

Above, the included \emph{abstractions} set common permissions for Bash scripts, among them access to \lstinline|.bash_profile|, \lstinline|.bash_rc| and \lstinline|.profile| files.

| The general rule of writing being in English, please correct "Figure 1. Struttura gerarchica dei profili" to "Figure 1. Hierarchical structure of profiles" (or similar).

Oops! We deserve Espresso Macchiato.

Changed to "Figure 1. Hierarchical structure of UserArmor profiles".

| On line 289 it appears written "32 GiB RAM". Most likely it is "32 GB RAM". On line 404 the same.

Done.

| About the "Bubblewrap" procedure, no details are provided at the first appearances (the first reference is to Figure 2 and line 292), but only at line 380. It is good that any name (consecrated or not) be presented at the first invocation.

Done.

| For CLINGO (some details about it), other possibilities can be mentioned and, why not, it would have been good to compare them with other ASP solutions.

We added an example and mentioned other ASP systems.
We mention the fact that solvers compiled for WebAssemly can run directly in the browser and therefore have no possibility for RCE.

| Figure 2 and Figure 3 are fine, but it's better to do more testing. For example, can UserArmor (since it derives from AppArmor which is implemented at the kernel level) be exploited by DoS? Vulnerabilities, depending on the kernel version, have allowed attackers to bypass AppArmor's protections.

The experiments measure the overhead introduced by UserArmor w.r.t. AppArmor. It seems to be minimal, so DoS should not be a problem. We commented on this aspect at the end of the introduction.

Round 2

Reviewer 1 Report

Comments and Suggestions for Authors

The authors have explained and corrected the issues from comments. The paper can be published now.

Reviewer 2 Report

Comments and Suggestions for Authors

The authors addressed all suggested issues. The article is fit for publication 

Author Response

Thank you! 

Reviewer 3 Report

Comments and Suggestions for Authors

Now the article is much clearer, easier to read. It's fine.

Author Response

Thanks!